# MgAl-NO₃ LDH: Adsorption Isotherms and Multivariate Optimization for Cr(VI) Removal

Anna Maria Cardinale [1],*[ID], Cristina Carbone [2][ID], Simone Molinari [3][ID], Gabriella Salviulo [3]
and Francisco Ardini [1][ID]

1    Department of Chemistry and Industrial Chemistry (DCCI), University of Genoa, Via Dodecaneso 31, 16146 Genoa, Italy
2    Department for the Earth, Environment and Life Sciences (DISTAV), University of Genoa, Corso Europa 26, 16132 Genoa, Italy
3    Department of Geosciences, University of Padua, Via Gradenigo 6, 35131 Padova, Italy
*    Correspondence: cardinal@chimica.unige.it; Tel.: +30-010-3356156

**Abstract:** Within the framework of the various strategies studied for the abatement of polluting agents in water, both from anthropogenic and natural origins, adsorption processes are among the most widespread techniques. In this context, Layered Double Hydroxides (LDHs) play a fundamental role. In this study, a Mg–Al LDH (nitrate intercalated, Mg/Al = 2) was prepared to be used as an anion exchanger for Cr(VI)-removal purposes from water. The LDH was synthesized through a co-precipitation reaction, followed by an aging process under heating. The compound was characterized by means of inductively coupled plasma–atomic emission spectroscopy (ICP-AES), X-ray powder diffraction (XRPD), field-emission scanning electron microscopy (FE-SEM) and Fourier-transform infrared spectroscopy (FT-IR). Regarding LDH adsorption capacity, with respect to Cr(VI), the adsorption isotherms and reaction kinetic were studied, and the adsorption process was well described by the Langmuir model. A central composite design was used for the multivariate optimization of the working parameters. The maximum adsorption capacity was estimated to be 30 mg/g.

**Keywords:** MgAl-NO₃ LDH; chromium (VI) removal; adsorption isotherm; experimental design

## 1. Introduction

Fresh water is essential for life on earth, while polluted water can have a massive impact on human health and on the environment. Because of the growing world population and contamination of water, freshwater scarcity has developed into a major global concern. Consequently, chromium discharge in the European Union waters is subjected to nationwide recommendations [1,2].

Contributing to this issue is the mismanagement of water distribution [3], pollution by agriculture and industrial development. This problem will most likely become more severe in the future due to the continuous growth in population and the growing industrialization of most countries in the world. Different organic and inorganic pollutants, from both natural and anthropogenic origin, can be found in water. In the framework of inorganic compounds, oxyanions are among the most common pollutants. Chromium (VI) compounds are extensively used in industrial processes, particularly in the engineering and chemical industries, such as in mining, electroplating facilities and the tannery industry [4]. Consequently, chromate contamination of water and soil should be closely monitored. In recent decades, great efforts have been made in both mitigating water pollution from industrial activities (i.e., through green technology) and in the purification of industrial wastewater. Regarding wastewater remediation, adsorption processes are widely used. They are carried out by means of adsorbents material, which should be effective but also as environmentally friendly as possible [5–12]. A potentially good class of adsorbents are layered double hydroxides (LDHs) due to their low cost, easy preparation procedures

and effectiveness. In addition, they show high chemical stability and, if properly planned as composition, are safe and environmentally sound. The crystal structure of LDHs is composed of brucite-type layers in which a trivalent cation partially substitutes a divalent cation, generating a net positive charge that is balanced by an anionic species in the interlayer, giving a general formula: $[M^{2+}_{(1-x)} M^{3+}_x (OH)_2]^{x+}(A^{z-})_{x/z} \cdot mH_2O$. Synthetic LDHs can be obtained through relatively simple routines of synthesis, combining a wide range of bivalent and trivalent cations with different ratios. Owing to their layered structure, LDHs can exchange the anion hosted in the interlayer and are known to have high anion-sorption capacities. For this reason, their performance as adsorbents has been the subject of various studies in recent years [13–18]. Among the known LDHs, Mg-Al-based LDHs have raised much interest as they are considered "green materials". The focus of this work is the synthesis of a MgAl LDH with nitrate as an anion, with a ratio Mg/Al = 2, to be used as a chromate anion adsorber, as well as the experimental study of adsorption isotherms. Moreover, the purpose of this study is to determine the efficiency of the adsorption as three chemical and physical parameters (LDH/volume of solution ratio, initial concentration Cr (VI), operating pH) vary independently in a large range of values. Knowledge of the pattern of LDH adsorption performance over a wide range of values of the three variables considered (each one independently of the others) is a great help in designing an effective adsorption process. This study requires many measures to correlate the adsorption efficiency to one parameter at a time.

The experimental design approach was used to optimize the planning and number of experimental measurements for the adsorption tests and to broaden the range of information available.

## 2. Materials and Methods

### 2.1. Chemicals

Chemicals were purchased and used without further treatment. The reagents $Al(NO_3)_3 \cdot 9H_2O$ (98.8% purity), $Mg(NO_3)_2 \cdot 9H_2O$ (98.9% purity) and NaOH (>98% purity) were supplied by VWR CHEMICALS, Leuven- Belgium. For the chromium-binding experiments, stock solutions of chromate (1000 mg $L^{-1}$) were prepared using potassium chromate ($K_2CrO_4$ > 99.0% purity) pro, 1,5-Diphenylcarbazide ($C_{13}H_{14}N_4O$), acetone and sulfuric acid ($H_2SO_4$, 95–98%), all provided by Sigma-Aldrich Co., LLC (St. Louis, MO, USA). All the solutions were prepared at the time of use with deionized water produced through a water purification system (M3/M6 Chemical Bürger s.a.s, Genova, Italy).

### 2.2. Synthesis and Characterization

The synthesis of the MgAl-nitrate LDH was carried out while maintaining a molar ratio of Mg/Al = 2. The stoichiometry of the compound results was $[Mg_{0.67}Al_{0.33} (OH)_2]$ $(NO_3)_{0.33} \cdot 3H_2O$. Different pathways of synthesis were tried and then, based on both the literature and on our previous experiences, the one chosen was that proposed by [19] during a coprecipitation reaction, followed by an aging process under heating. The deionized water used in each synthesis was decarbonated on the spot to minimize the $CO_2$ content, and the entire process was carried out under an argon atmosphere to ensure the presence of the nitrate ion in the interlayer of the structure, avoiding the carbonate anion in the compound. A proper amount of the magnesium and aluminum nitrates, based on the desired Mg/Al ratio, was dissolved in 200 mL of boiled deionized water under an argon atmosphere. Under a controlled atmosphere, a NaOH 1 M solution (also prepared in decarbonated deionized water) was slowly added dropwise into the nitrate solution under magnetic stirring until the pH value of the suspension was over pH = 9. The whole mixture was then transferred to a dark bottle and aged in a stove at 70 °C for twelve hours. After aging, the obtained white compound was separated from the solution by centrifugation at 7000 rpm for 10 min, repeatedly washed with water, and dried in a stove at 60 °C for 24 h. The yield of the processes was about 80%.

After all syntheses and before applying them in the adsorption test, the samples were characterized by means of several techniques as follows: inductively coupled plasma–atomic emission spectroscopy (ICP-AES); X-ray powder diffraction (XRPD); field-emission scanning electron microscope (FE-SEM); and Fourier-transform infrared spectroscopy (FT-IR). The ICP-AES measurements were performed using an axially viewed Varian (Springvale, Australia) Vista PRO. The sample introduction system consisted of a glass concentric K-style pneumatic nebulizer (Varian) joined to a glass cyclonic spray chamber (Varian). In order to compensate for non-spectral interferences, on-line internal standardization (4 µg mL$^{-1}$ Lu standard solution) was applied. XRPD patterns were collected to identify the eventual crystalline phase using an X'Pert MPD (Philips, Almelo, Netherland) X-ray powder diffractometers were equipped with a Cu anticathode with Cu Kα1 radiation (λ = 1.5406 Å), and the samples were prepared by grinding them in an agate mortar. The patterns were collected between 10° and 90° 2θ with a step of 0.001° and a measuring time of 50 s/step.

The indexing of the obtained diffraction data was performed by comparison with spectra available in the literature [20], and the lattice parameters of the phases were calculated using the LATCON program [21]. A ZEISS SUPRA 40 V model of the field-emission scanning electron microscope (FE-SEM) was used, in which the sample was analyzed by applying an acceleration voltage of 5 kV for 50 s. The FT-IR spectra were collected in the usual wavelength range of 4000 to 600 cm$^{-1}$ using a Spectrum 65 FT-IR Spectrometer (PerkinElmer, Waltham, MA, USA) equipped with a KBr beam-splitter and a DTGS detector using an ATR accessory with a diamond crystal. The residual chromium concentration in the solution was determined by means of UV-VIS spectrophotometry (instrument Varian Cary 50 Scan) using the 1,5-diphenylcarbazide method [22].

*2.3. Adsorption Tests*

For the experimental design, the LDH and the chromate solution were kept in batch equilibration in a rotary shaker at 15 rpm for 30 min. To evaluate the chromium adsorption kinetics and establish the minimum efficient contact time, different solutions with 80 ppm of Cr(VI) were left with 10 g/L of LDH in batch equilibration for increasing times ranging from 1 to 90 min, and were then analyzed. This test showed that equilibrium is reached very quickly, and a contact time of 30 min was chosen to ensure both equilibrium and the standardization of the recovering procedure were achieved. Then, the solid was separated from the liquid phase during centrifugation at 5000 rpm, dried in an oven at 70 °C and finally stored in a desiccator. To determine the residual chromate concentration in the solution, after the adsorption process, the samples were analyzed by UV-VIS spectrophotometry. Data were processed using the open-source software environment for statistical computing and graphics 'R' [23] with the additional package CAT (Chemometric Agile Tool) [24]. In particular, the software performed multiple linear regression, in accordance with the ordinary least-squares method and was able to build isoresponse plots and response surfaces deriving from the computed models [25].

The adsorption tests to evaluate chromium removal efficiency at extreme pH values were carried out using 100 ppm Cr(VI) solutions, keeping the LDH mass constant at 10 g L$^{-1}$ and varying the pH of the chromium solution from 2 to 14 by adding HNO$_3$ or NaOH. The pH value did not assume values lower than pH ≈ 2 as the LDH could have degraded.

Finally, the equilibrium adsorption of chromium (VI) on the LDH was investigated with a series of batch experiments. Known concentrations of Cr(VI) (25–300 mg L$^{-1}$) were added to LDHs (10.0 g L$^{-1}$) and the systems were mixed in a rotary shaker for 30 min. After equilibrium accomplishment, LDHs were separated and the residual Cr(VI) concentration in the supernatant was quantified by UV-Vis. The adsorption process can be represented as follows: $CrO_4^{2-}{}_{(aq)} + MgAl-NO_{3(solid)} \rightleftarrows 2 NO_3^-{}_{(aq)} + MgAlCrO_{4(solid)}$.

In this case, one mole of chromate (divalent) replaced two moles of nitrate (monovalent) in the LDH lattice. Furthermore, the adsorption capacity depends mainly on

interlayer exchange, but other adsorption mechanisms could occur even if they are of lesser importance.

## 3. Results and Discussion

### *3.1. LDH Characterization*

After synthesis, the obtained compound was analyzed by ICP-AES analysis and the stoichiometric ratio between the two Mg and aluminum cations was confirmed as the desired 2/1 ratio. Regarding structural characterization, in Figure 1 the XRPD pattern (showing the typical turbostratic feature); main typical symmetric reflections at (003) = 11.6°, (006) = 23.3°, (009) = 34.6°, (110) = 60.8° and (113) = 62.1°; and asymmetric (015) = 38.7°, (018) = 46.0° reflections are present. The small signal at about 20°, partially superimposed on reflection (006), can be attributed to traces of magnesium hydroxide impurities. The compound pertains to the R-3 m h space group and the cell parameters were calculated with the following results: a = 0.3040(6) nm; c = 2.2846(9) nm; and Vcell = 0.183(1) nm$^3$. The brucite layer thickness was evaluated to be approximately 0.24 nm [26], so the interlayer regions thickness can be estimated to be 0.52 nm.

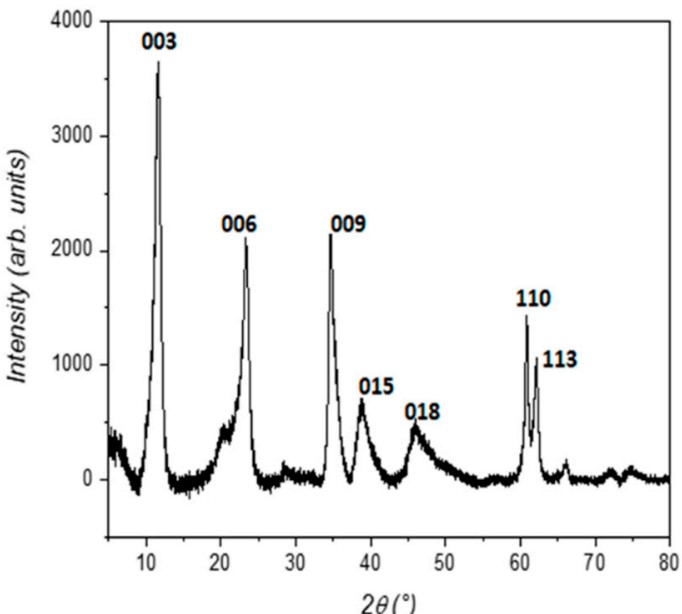

**Figure 1.** Powder X-ray diffraction pattern of the LDH synthesized.

With regard to morphology, Figure 2 reveals a homogeneous structure, which tends to transform into lamellae. The typical flower-like lamellar organization was not detected in the analyzed samples. This shape is the usual assumed shape of the compounds obtained through coprecipitation synthesis followed here [27]. EDX analysis of the sample further confirmed the ratio of the two metals in the compound.

Figure 3 shows the infrared spectrum of the MgAl LDH. The broad adsorption band at 3420 cm$^{-1}$ is the characteristic signal of the O-H bond stretching of the water molecules and of the hydroxyl groups present in the interlayers of the structure, while the one at 1638 cm$^{-1}$ is related to H-O-H bending. The peak at 1357 cm$^{-1}$ is related to the vibration of the nitrate ion in the interlayer. Furthermore, the signal at 767 cm$^{-1}$ can be attributed to the nitrate ion. Additionally, this band falls in the region of the O-M-O stretching absorption.

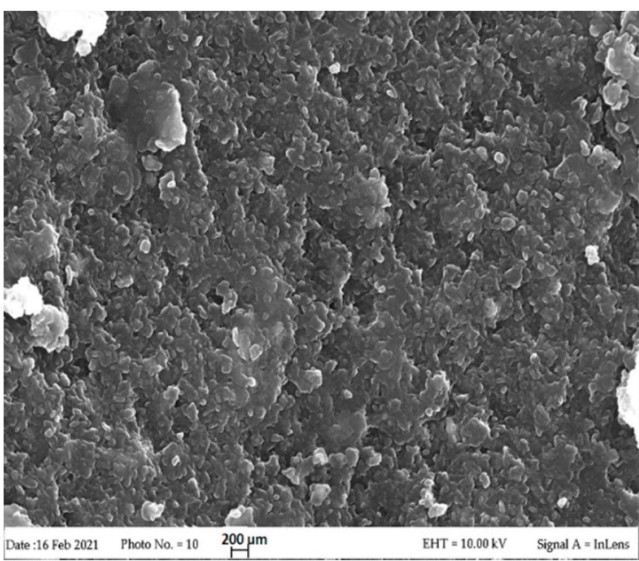

**Figure 2.** FE-SEM image of the LDH synthesized.

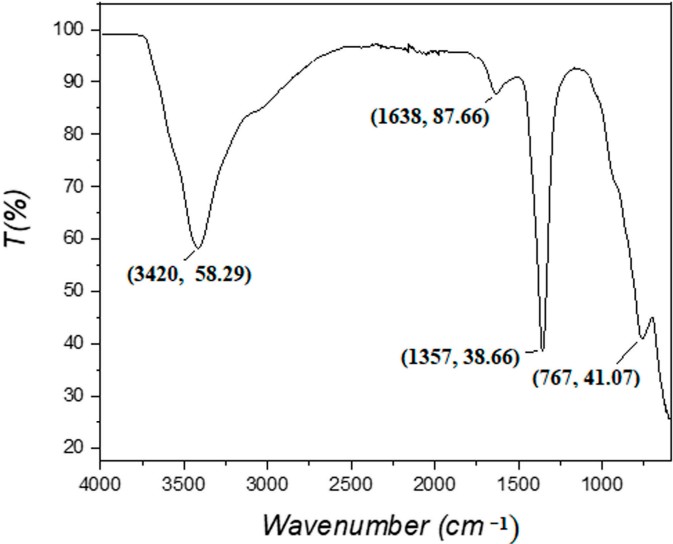

**Figure 3.** IR spectrum of the LDH synthesized.

### 3.2. Adsorption Tests

In order to study the adsorption characteristics and performance of the LDHs with respect to Cr(VI), different parameters had to be evaluated, namely the ratio mass of the LDH/solution volume and the pH and the concentration of Cr(VI) in the solution. To obtain the best knowledge of the system, a decision was made to apply an experimental design (or design of experiments, DOE). This is a multivariate strategy that is more convenient than the classic univariate approach because it requires fewer experimental runs; moreover, it is able to detect interactions among the variables and it yields empirical mathematical relationships between variables and responses [28]. After the selection of the variables (or factors), the experimental domain was assessed, i.e., the minimum and maximum values of the considered factors. Chromium concentrations were set between 100 and 300 mgL$^{-1}$, whereas values for pH were investigated between 4 and 8. For the values of the ratio LDH mass/solution volume, a preliminary screening study was performed on a solution at 80 ppm Cr(VI) in order to evaluate the variation of adsorbed Cr(VI) related to the LDH mass/solution volume. Figure 4 shows the percentage of chromium adsorbed with respect to the mass of the LDH as gL$^{-1}$. The ratio considered optimal corresponds to values close

to 8–10 gL$^{-1}$ of the absorbent. Therefore, the extreme values for the experimental design were chosen as 5 and 15 gL$^{-1}$.

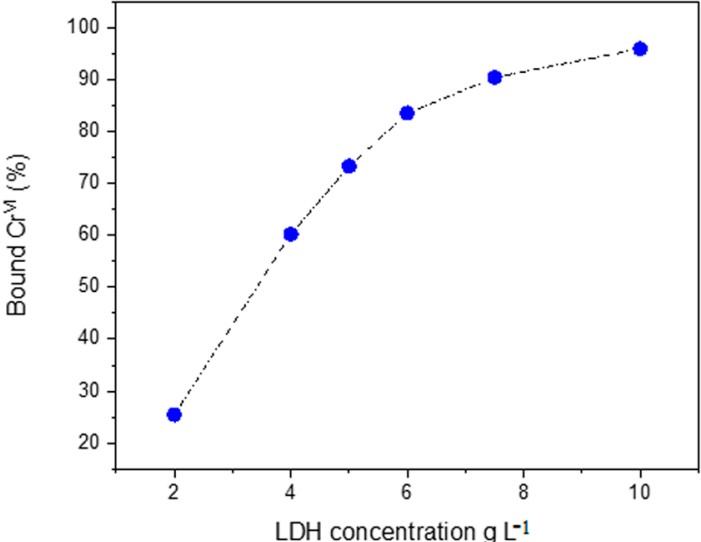

**Figure 4.** Percentage of adsorbed chromium vs. the LDH concentration in an 80 ppm C(VI) solution.

The selected design was the face-centered central composite design (CCD), which explores the factors at three levels [28]. A total of 34 experiments were performed, consisting of two distinct replicates of 14 conditions and 6 replicates of condition 15, i.e., the center point (Table 1).

**Table 1.** Experimental conditions of the adsorption tests performed.

| Experiment | Ratio (g/L) | pH | C (ppm) |
| --- | --- | --- | --- |
| 1 | 5 | 4 | 100 |
| 2 | 15 | 4 | 100 |
| 3 | 5 | 8 | 100 |
| 4 | 15 | 8 | 100 |
| 5 | 5 | 4 | 300 |
| 6 | 15 | 4 | 300 |
| 7 | 5 | 8 | 300 |
| 8 | 15 | 8 | 300 |
| 9 | 15 | 6 | 200 |
| 10 | 10 | 8 | 200 |
| 11 | 10 | 6 | 300 |
| 12 | 5 | 6 | 200 |
| 13 | 10 | 4 | 200 |
| 14 | 10 | 6 | 100 |
| 15 | 10 | 6 | 200 |

To minimize the influence of systematic trends, the experiments were performed randomly; the only exception were the six replicates of the center point, which were carried out regularly throughout the sequence of analysis to give the estimate of the experimental variance, which was necessary to evaluate the significance of the models' coefficients.

The obtained data were then subjected to tests of multiple regression using CAT software (see the Section 2), obtaining a mathematical model able to describe the percentage of adsorbed Cr(VI) as a function of the three studied factors in the domain of interest (Equation (1)):

$$Cr(VI)\% = 100.3247 + \mathbf{3.6895 \cdot R} + 0.2135 \cdot pH - \mathbf{2.3280 \cdot C} - 0.2000 \cdot R \cdot pH + \mathbf{2.9525 \cdot R \cdot C} - \mathit{0.9600 \cdot pH \cdot C} \\ - 2.5620 \cdot R^2 - 1.0870 \cdot pH^2 - 1.0945 \cdot C^2 \tag{1}$$

where R, pH and C are the three factors, i.e., ratio LDH mass/solution volume, pH and concentration of Cr(VI), respectively. Coefficients which were significant at a 95% confidence level are highlighted in italic, whereas those significant at a 99.9% confidence level are highlighted in bold. The coefficients are also reported in Figure 5a.

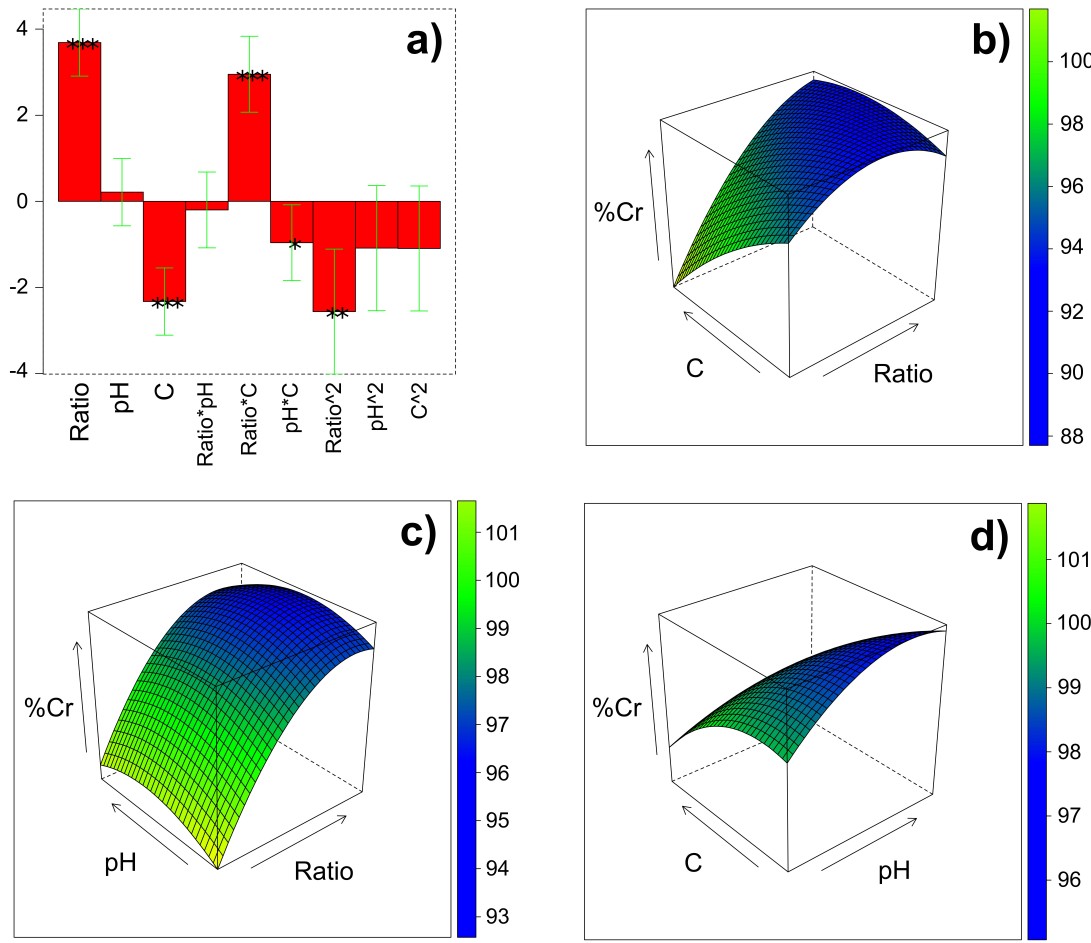

**Figure 5.** (**a**) Significance of the coefficients. (**b**–**d**) Response surface plots for binary influence: (**b**) Cr(VI) concentration ratio and LDH mass/solution volume; (**c**) pH ratio and LDH mass/solution volume; (**d**) Cr(VI) concentration and pH (* = 95% confidence level; ** = 99% confidence level; *** = 99.9% confidence level).

The two variable LDH mass/volume and solution chromium concentrations showed a strong positive and negative influence on the response, respectively. This means that the percentage of adsorbed Cr(VI) increases by increasing the LDH mass/volume and by decreasing the Cr concentration. On the other hand, pH proved to have few significant effects on the response in this experimental domain, except for its interaction with Cr concentration. The entire behavior of the system can be fully appreciated by looking at the three-dimensional response surfaces for all the combinations of pairs of parameters vs. the

response, which are reported in Figure 5b–d. It can be seen that the percentage of Cr(VI) adsorbed is very high in all domains (always above 88%, mostly above 95%), with maximum values corresponding to high LDH/solution volume ratios and low concentrations of Cr(VI). The effect of pH shows a slight but statistically significant increase at low Cr concentrations, although the improvement is almost irrelevant from a practical point of view (from 99% to 102%). This model shows the high performances of Cr adsorption for these LDHs in these conditions, allowing a prediction of its capabilities within the whole experimental domain, including conditions that were not directly tested as experiments. The following step in this study consists of the behavior of these LDHs outside the investigated experimental domain, considering more extreme conditions of pH and Cr concentration. According with the tests previously carried out, the LDH mass/volume was set to 10 g/L because it represents a good compromise between yield optimization and minimum compound consumption. The pH varied from 3 to 14 using a solution of 100 ppm of Cr (Figure 6). It can be seen that the adsorption is almost complete in extremely acidic, neutral and slightly basic pH conditions; from pH $\approx$ 11, the percentage of chromium absorbed rapidly decreases until it reaches about 20% at pH $\approx$ 14.

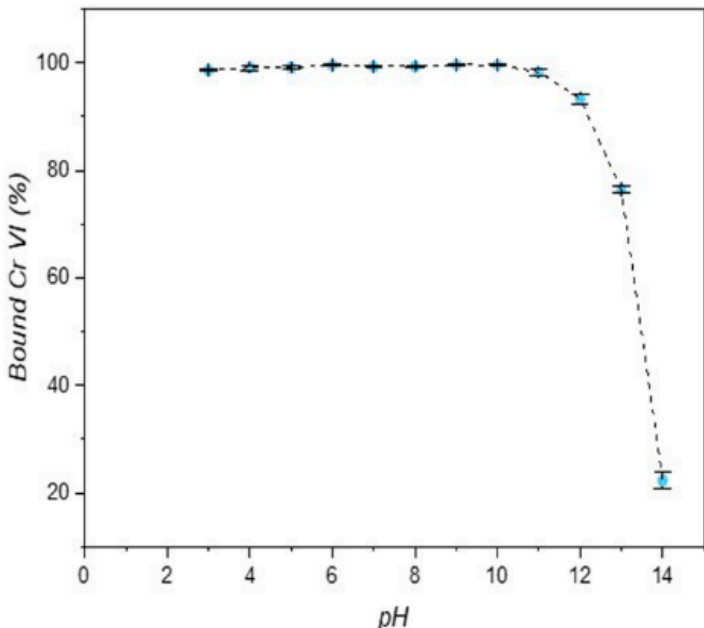

**Figure 6.** Trend in Cr(VI) adsorption capability vs. pH.

This may be due to the high amount of hydroxyl anion at very high pH value, which competes with all other anions in the solution for the interlayer. Although the hydroxyl anion shows low LDH affinity, the large concentration in the solution (at high pH values) can shift the equilibrium in its favor. In Figure 7, the PXRD patterns for the pristine LDH and the LDH after batch equilibration with a sample at 100 ppm Cr(VI) at two different pH values, 3 and 13, respectively, are compared. The plot clearly shows lower crystallinity for the sample at pH = 3, the peaks are less resolved and, specifically, the separation between the signal of the reflection (006) of the LDH at 23.3° and that of the Mg(OH)$_2$ impurity at 20° is evident. It can be hypothesized that at a particularly acidic pH, the LDH structure tends to collapse, and this justifies the loss of crystallinity. The XRD spectrum of the sample after adsorption at pH = 13, on the other hand, seems to show an improvement in the crystallinity degree of the structure and an increase in the intensity of the peaks compared to the pre-adsorption LDH. No shifting in peak position occurred after the incorporation of the chromate ion.

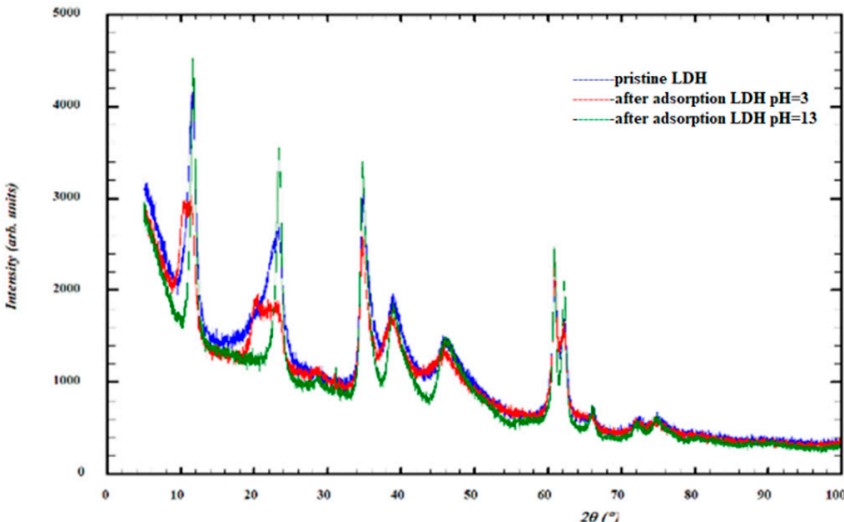

**Figure 7.** PXRD patterns of the pristine MgAl-NO$_3$ LDH (blue line) after Cr(VI) extraction at pH = 3 (red line) and pH = 13 (green line).

As a preliminary experiment, the adsorption kinetics were evaluated. The results are shown in Figure 8a,b where qt and qe are, respectively, the mass of chromium absorbed at time t and at equilibrium. The experimental parameters are well described by a pseudo second-order equation where linear and non-linear forms are described by Equations (2) and (3), and k is the adsorption constant [29,30].

$$q_t = \left(kq_e^2t\right)\Big/ 1 + kq_et \qquad (2)$$

$$t\Big/q_t = 1\Big/\left(kq_e^2\right) + 1\Big/q_et \qquad (3)$$

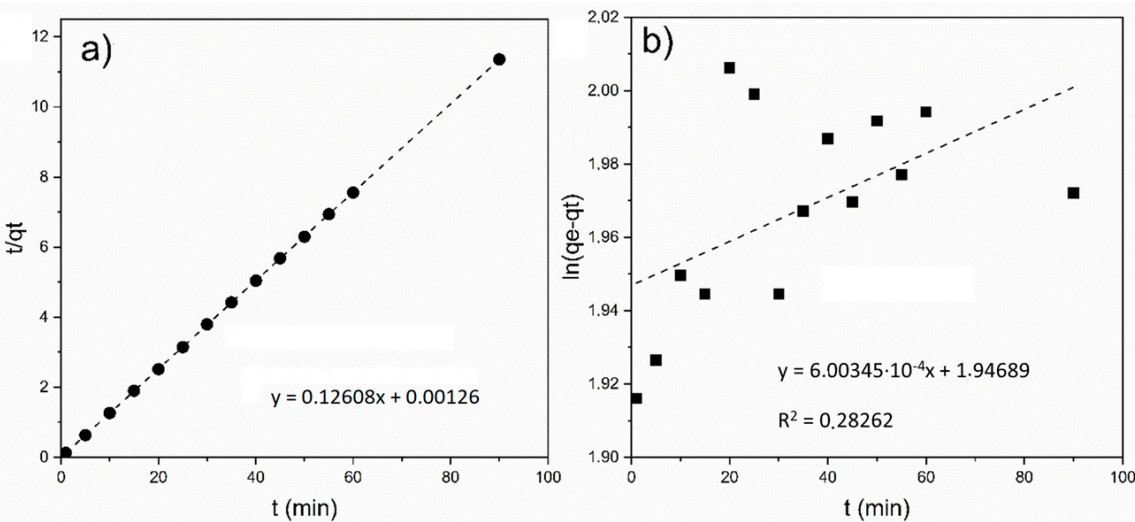

**Figure 8.** Binding kinetics of Cr$^{VI}$ adsorption onto MgAl-NO$_3$ LDH. In (**a**), the experimental data are fitted with a pseudo second-order kinetic model, whilst in (**b**), the experimental data are fitted with a pseudo first-order kinetic model.

It can be seen that after a few minutes of contact time, the adsorption process can be considered complete. For this reason, the batch equilibration time selected was 30 min to guarantee a standard LDH recovery procedure. Moreover, the binding behavior of Cr(VI) to LDHs was further investigated at equilibrium according to the Langmuir [31] isotherm model.

The Langmuir equation was applied In its linearized form (Equation (4)):

$$C_e \Big/ Q_e = 1 \Big/ (q_m K_L) + C_e \Big/ q_m \tag{4}$$

where $C_e$ is the equilibrium concentration of Cr(VI), $q_m$ is the maximum amount of bound Cr(VI) and $K_L$ is the stability constant. The values of $q_m$ and $K_L$ were quantified from the slope and the intercept of the linear plot of $C_e/Q_e$ vs. $C_e$. The Langmuir model well fitted the experimental data (see Figure 9) and saturation-binding behavior of LDHs was observed. The maximum adsorption capacity was 27.397 mg/g at neutral conditions.

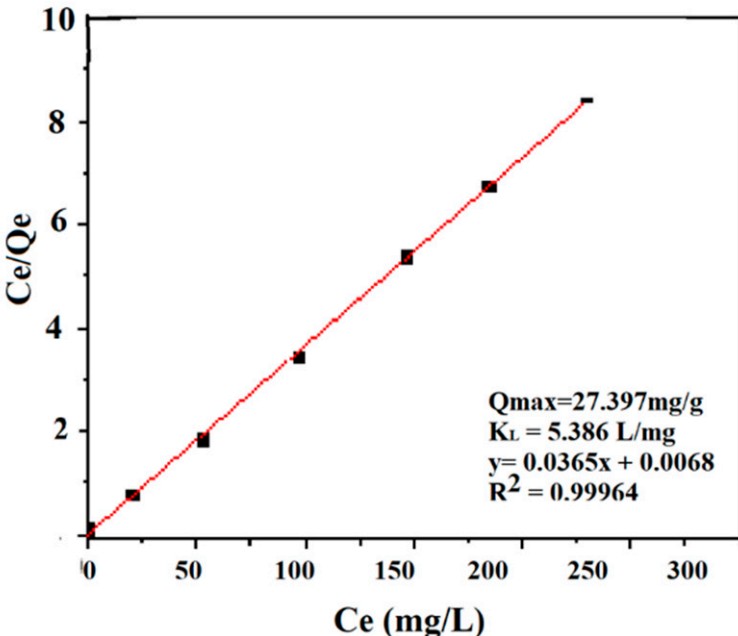

**Figure 9.** Linear plot of Langmuir isotherm model for Cr(VI) adsorption onto LDHs.

A great number of compounds and/or materials have been studied as pollutant adsorbents in wastewater. Focusing on the LDHs and Cr(VI) to compare adsorption performance among different compounds, it is necessary take into account a wide number of parameters (e.g., formula, solution concentration of Cr(VI), operating conditions, co-presence of other pollutants). In Table 2, a comparison of the adsorption behavior of the studied compound is reported, along with other similar LDH materials previously studied. The comparison was conducted between MgAl-based LDHs or LDHs constituted to another metal other than Mg and Al, and has been performed among processes occurred at room temperature. Owing to the literature data available, the study was between a nitrate LDH (this work) and other carbonate LDHs. The nitrate (monovalent ion) was less affine than the carbonate (divalent anion) in the interlayer. The NiMgAl compound appeared to show the best performance, and the XRD pattern for this ternary LDH showed an interlayer larger than the nitrate under investigation, facilitating the entry of a chromate ion bulkier than both nitrate and carbonate.

**Table 2.** Adsorption capacity of MgAl-LDHs for Cr(VI).

| LDH Composition | Solid/Liquid Ratio (g/L) | Solution Cr(VI) Concentration (ppm) | $q_e$ (mg/g) | Ref. |
|---|---|---|---|---|
| MgAl-NO$_3$ | 10 | 300 | 27.397 | This work |
| MgAl-CO$_3$ * | 5 | 87 | 2.37 | [32] |
| NiMgAl-CO$_3$ | 0.5 | 100 | 32.5 | [33] |
| MgAl-CO$_3$ | 1 | 100 | 11.55 | [18] |

* Sample tested on a real industrial wastewater. The Cr(VI) adsorbed does not represent the $q_e$ value.

## 4. Conclusions

A MgAl-nitrate LDH was synthesized and tested as a Cr(VI) adsorbent for remediation purposes. Before the adsorption tests, the synthesized compound was characterized. The chemical physical interactions between the LDH and the chromium (VI) solution, together with the reaction kinetic, were investigated, correlating the different variables involved. For this purpose, the experimental design was used to optimize the setting of the experimental measures. From the experimental results, the following conclusions can be drawn:

- The synthesis of MgAl-nitrate LDH is simple, reproducible and provides good yield.
- To plan the experimental setup, applying an experimental design (DOE) dramatically reduces the number of experimental tests and yields an appreciation of the whole behavior of the system.
- The ratio (mass of LDH)/(solution volume) identified for optimal adsorption of chromium corresponds to 10 gL$^{-1}$. Foreseeing a large-scale use of the adsorbent, this value represents the better compromise between adsorption capability and the cost effectiveness of the process.
- As the kinetic, it has demonstrated an almost instantaneous removal of chromium. The Langmuir isotherm adequately describes the absorption kinetics.
- Among the different adsorption parameters investigated, the solution's pH value has no influence on the efficiency of the process, as long as the pH is between 3 and 12. As predicted, the chromium adsorption increases for high LDH/solution volume ratios and low concentrations of Cr(VI).The maximum adsorption capacity is 27.397 mg/g at neutral conditions.
- The percentage of chromium absorbed (with a ratio of 10 gL$^{-1}$ of LDH) under the studied conditions is 100 mass% up to an initial concentration of about 300 ppm. This performance can meet remediation needs in the majority of cases where it could be applied.
- By comparing the performance of the investigated MgAl nitrate with other similar LDHs operating in similar chemical—physical conditions, it seems that the nitrate anion can be exchanged more easily than the carbonate. Only the ternary NiMgAl LDH shows a better adsorption capacity.

**Author Contributions:** Conceptualization, A.M.C., C.C. and G.S.; Methodology, S.M.; Software, F.A.; Data curation, S.M., G.S. and F.A.; Writing—original draft, A.M.C.; Writing—review & editing, S.M., G.S. and F.A.; Supervision, A.M.C. All authors have read and agreed to the published version of the manuscript.

**Funding:** This research received no external funding.

**Data Availability Statement:** The data presented in this study are available on request from the corresponding author.

**Acknowledgments:** We want to thank Walter Sgroi for the help provided in the UV-VIS spectroscopy measurements.

**Conflicts of Interest:** The authors declare no conflict of interest.

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
