# Peer review of "MgAl-NO3 LDH: Adsorption Isotherms and Multivariate Optimization for Cr(VI) Removal"

_chemistry, doi:10.3390/chemistry5010045_

Round 1

Reviewer 1 Report

Comments to authors

1/ The nouvelty of this work is not clearly stated in the introduction section. Could you please provide more information on what makes your work novel and add this information in the introduction section to provide context for your results?

2/ Can you provide more information on the background of the problem of chromate contamination in water and soil and why it is an issue of global concern?

3/ Could you elaborate on the purpose of the preliminary screening study performed to determine the optimal ratio mass of LDH/solution volume, and how it informed the selected range for this factor in the CCD?

4/ Increase the resolution of the images to ensure that the details are visible, and the figures are not blurry or pixelated.

5/ Can you describe the chemical reactions that occur during the adsorption of chromium onto layered double hydroxides?

6/ Manuscripts should refer to and cite as much as possible from the last five years. Some high-quality literatures about sustainability of water in recent years can be referenced and cited in the introduction section, such as Arabian Journal for Science and Engineering 1-11 (2021) https://doi.org/10.1007/s13369-022-06899-y ; Journal of Desalination and Water Treatment 194 (2020) 439-449.

Author Response

  1. 1 Comments to authors

Thanks for the fruitful suggestions, in the following the answers.

1/ The nouvelty of this work is not clearly stated in the introduction section. Could you please provide more information on what makes your work novel and add this information in the introduction section to provide context for your results?

Answer: the introduction section has been improved following the suggestion.

2/ Can you provide more information on the background of the problem of chromate contamination in water and soil and why it is an issue of global concern?

Answer: the introduction section has been improved following the suggestion.

3/ Could you elaborate on the purpose of the preliminary screening study performed to determine the optimal ratio mass of LDH/solution volume, and how it informed the selected range for this factor in the CCD?

Answer: the tests leading to the choice of the solid/liquid ratio have been added to the experimental section

4/ Increase the resolution of the images to ensure that the details are visible, and the figures are not blurry or pixelated.

Answer: the images have been formatted at a resolution of 300dpi, as required from the instructions for the authors.

5/ Can you describe the chemical reactions that occur during the adsorption of chromium onto layered double hydroxides?

Answer-The adsorption process doesn’t involve properly a chemical reaction, it has been described in section 2.3

6/ Manuscripts should refer to and cite as much as possible from the last five years. Some high-quality literatures about sustainability of water in recent years can be referenced and cited in the introduction section, such as Arabian Journal for Science and Engineering 1-11 (2021) https://doi.org/10.1007/s13369-022-06899-y,   Desalination and Water Treatment, 2020, 194, pp.439 - 49. âŸ¨10.5004/dwt.2020.25562⟩. âŸ¨hal-03044363⟩

Answer: the two suggested references have been added.

Reviewer 2 Report

This work describes the development of Mg-Al layered double hydroxide by aqueous precipitation, its characterization and evaluation as an adsorbent for Cr(VI) uptake from water. The subject of this paper has potential interest though its novelty is limited. A significant problem of the manuscript is the very low level of English use and the weakness in the presentation and interpretation of obtained results. I would not recommend further consideration of this work. Here are some more detailed comments:

There severe issues in the linguistic part of the manuscript. It must be checked and corrected thoroughly by a fluent English speaker

Line 40. Adsorbents not absorbers

The duration of adsorption tests is very small. I don’t believe 30 minutes are sufficient to reach equilibrium.

Line 115. Please explain the data processing with these software.

Line 131. (003)=11.6o this is not a proper way to show these identification.

Line 153. Peak at 767 cm-1 is attributed to nitrate ions. No explanation is given on that although this region with possible contributions from Mg-O and Al-O bands.

Author Response

R.2

Thanks for the fruitful suggestions, in the following the answers.

This work describes the development of Mg-Al layered double hydroxide by aqueous precipitation, its characterization and evaluation as an adsorbent for Cr(VI) uptake from water. The subject of this paper has potential interest though its novelty is limited. A significant problem of the manuscript is the very low level of English use and the weakness in the presentation and interpretation of obtained results. I would not recommend further consideration of this work. Here are some more detailed comments:

-There severe issues in the linguistic part of the manuscript. It must be checked and corrected thoroughly by a fluent English speaker

Answer: the text has been improved and the language has been checked

-Line 40. Adsorbents not absorbers ok

Answer: the word has been corrected.

-The duration of adsorption tests is very small. I don’t believe 30 minutes are sufficient to reach equilibrium.

Answer: the duration of the batch equilibration process has been chose on the basis of the results that can be seen in figure 8a), as described in the text.

-Line 115. Please explain the data processing with these software .

Answer: The software is free and the scripts are available at the link indicated in the reference [23]. The general operations have been added with the following sentence in section 2.3

-Line 131. (003)=11.6o this is not a proper way to show these identification.

Answer: I apologize for the typo, it has been corrected

-Line 153. Peak at 767 cm-1 is attributed to nitrate ions. No explanation is given on that although this region with possible contributions from Mg-O and Al-O bands.

Answer: The IR spectrometry has been performed mainly to check the effectiveness of the compound obtained, anyway the discussion of the band has been implemented.

Reviewer 3 Report

In this study, a Mg–Al LDH (nitrate intercalated, Mg/Al =2) was prepared, and was used as anion exchanger for Cr(VI) removal from water. The LDH was synthesized throughout a coprecipitation reaction, followed by an aging process under heating. The study is interesting, and fall in the scope of the journal, However, the manuscript lacks of novelty and there are still several questions to be answered.

1.There are lots of published paper in Cr(VI) removal by novel material, such as Sun et al., Bioresource Technology, 2023, 369, 128373. Wang et al., Separation and Purification Technology, 2023, 306, 122631.  etc, More research papers should be compared and analyzed in the introduction. The novelty of the paper should be emphasized.

2. Why does the author directly set the ratio of Mg/Al=2. Please give the reason.

3. The author should add another one table of comparison of the present study with previous study to evaluate the effectiveness of Mg–Al LDH.

4. The format of the references is not uniform, please further check and revise.

5. Line 132: As the characterization, in figure 1 is reported the XRPD pattern (showing the typical turbostratic feature), the main typical symmetric reflections at (003)=11.6°, (006)=23.3°,(009)=34.6°, (110) =60.8°, (113)=62.1 and asymmetric (015)=38.7, (018)=46.0° reflections are present. Adding the ° in the corresponding position.

6. Line 153: “cm-1” should be changed to “cm-1

7. The format of the manuscript should be in accordance with the guide for author of the Journal. Please revise the manuscript carefully.

Author Response

R.3

Thanks for the fruitful suggestions, in the following the answers.

In this study, a Mg–Al LDH (nitrate intercalated, Mg/Al =2) was prepared, and was used as anion exchanger for Cr(VI) removal from water. The LDH was synthesized throughout a coprecipitation reaction, followed by an aging process under heating. The study is interesting, and fall in the scope of the journal, However, the manuscript lacks of novelty and there are still several questions to be answered.

1.There are lots of published paper in Cr(VI) removal by novel material, such as Sun et al., Bioresource Technology, 2023, 369, 128373. Wang et al., Separation and Purification Technology, 2023, 306, 122631.  etc, More research papers should be compared and analyzed in the introduction. The novelty of the paper should be emphasized.

Answer:  the introduction has been improved as suggested.

  1. Why does the author directly set the ratio of Mg/Al=2. Please give the reason.

Answer: different ratio between Mg and Al have been applied to the MgAl based LDH synthesis, the ratio mainly affects the crystallinity of the compound. In this work the ratio 2 has been chose to compare, in a subsequent step, the behavior in real wastewater with another MgAl LDH with carbonate as anion previously sinthesised [32].

  1. The author should add another one table of comparison of the present study with previous study to evaluate the effectiveness of Mg–Al LDH.

Answer: a comparison among different LDHs has been added in the discussion

  1. The format of the references is not uniform, please further check and revise.

-Answer :The format has been checked following the instruction for authors

  1. Line 132: ‘As the characterization, in figure 1 is reported the XRPD pattern (showing the typical turbostratic feature), the main typical symmetric reflections at (003)=11.6°, (006)=23.3°,(009)=34.6°, (110) =60.8°, (113)=62.1 and asymmetric (015)=38.7, (018)=46.0° reflections are present.’ Adding the ‘°’ in the corresponding position.

Answer: I apologize for the typo, that have been corrected

  1. Line 153: “cm-1” should be changed to “cm-1

Answer: I apologize for the typo, that have been corrected

  1. The format of the manuscript should be in accordance with the guide for author of the Journal. Please revise the manuscript carefully.

Answer: the format has been revised.

Round 2

Reviewer 1 Report

No comments

Reviewer 2 Report

Authors effort improved the manuscript a lot and can be published although the novelty level of the paper is not at the highest level.

Reviewer 3 Report

The comments have been answered. It can be accept in the present form.